# Benchmarking Multimodal Clinical Foundation Models to Reveal Significant Demographic Disparities

**Milan Mohan** [1]   **Saketh Lingisetty** [1]   **Umar Ahmad** [1]   **Avi Kumar** [1]   **Shankar Harikrishnan** [1]   **Sanjana Chamarty** [1]   **Kevin Zhu** [1]

## Abstract

Medical imaging foundation models produce high-dimensional structured feature vectors for downstream tabular classifiers, bridging visual pretraining and structured clinical data pipelines. We benchmark MedImageInsight, MedSigLIP, and BiomedCLIP on INSPECT, pairing CTPA imaging with longitudinal EHR, using linear and MLP probes across PE diagnostic and seven prognostic tasks. We report three findings. First, logistic regression outperforms all MLP variants for MedImageInsight ($+0.014$) and MedSigLIP ($+0.021$), suggesting approximately linearly separable structure in high-capacity embeddings. Second, structured EHR adds at most $+0.003$ AUROC over frozen CT alone ($+0.017$ for BiomedCLIP readmission), suggesting simple early fusion cannot compensate for representational deficits. Third and primarily, age is the dominant disparity: patients aged 18–40 face underdiagnosis rates (UDR) of 0.63–0.80 versus 0.31–0.41 for ages 75–90 (gap 0.32–0.45), exceeding race/ethnicity and gender gaps across all eight tasks. Adversarial debiasing is the only mitigation that reduces gaps without hurting AUROC: age-targeted debiasing cuts MedImageInsight's gap by 79% ($0.333 \rightarrow 0.069$; $p < 0.001$) at 0.011 AUROC cost. BiomedCLIP debiasing is unreliable ($p \geq 0.12$), suggesting embedding expressivity may be a prerequisite for demographic disentanglement.

Accepted at the Workshop on Foundational Models for Structured Data at ICML 2026. [1]Algoverse AI Research. Correspondence to: Kevin Zhu <kevin@algoverseairesearch.org>.

*Proceedings of the $43^{rd}$ International Conference on Machine Learning*, Seoul, South Korea. PMLR 306, 2026. Copyright 2026 by the author(s).

## 1. Introduction

A key question for foundation models applied to structured data is how representations learned from large-scale pretraining transfer to downstream tabular prediction tasks. Medical imaging foundation models offer a concrete test case: their output embeddings, fixed-length, high-dimensional vectors, can be treated as structured feature inputs to any tabular classifier, creating a general-purpose pipeline from visual pretraining to structured clinical prediction without task-specific fine-tuning.

The INSPECT dataset (Huang et al., 2023) provides an ideal benchmark: 23,248 CTPA studies linked to structured longitudinal EHR records across eight PE diagnostic and prognostic tasks. PE is a life-threatening condition where missed diagnoses directly worsen outcomes (Mayo Clinic Staff, 2022; Walter, 2023), making this evaluation clinically consequential. Rather than establish state-of-the-art diagnostic performance, we aim to characterize these embeddings as structured data artifacts.

We benchmark three frozen encoders as structured feature extractors across eight clinical prediction tasks, revealing that high-capacity encoder embeddings exhibit approximately linear task-relevant structure with direct implications for structured data pipeline design. We then conduct the first systematic fairness audit of these structured representations across race/ethnicity, gender, and age, revealing age as the dominant and previously unreported disparity dimension on INSPECT across all tasks. Lastly, we evaluate adversarial debiasing as a post-processing strategy for FM-derived structured pipelines, identifying encoder capacity as a prerequisite for reliable demographic disentanglement.

## 2. Related Work

INSPECT (Huang et al., 2023) connects CTPA imaging with EHR for PE diagnosis and prognosis, providing Long-term Recurrent Convolutional Network (CT-LRCN) baselines, to evaluate frozen FM embeddings. Recent work has increasingly explored foundation model embeddings for multimodal clinical data integration (Menon et al., 2026; Amar et al., 2025), as well as feature extraction across pathology,

time-series, and genetics (Cui et al., 2026; Lee et al., 2024; Mukherjee et al., 2024). Moreover, the strong performance of linear probes aligns with prior observations that simple logistic regression often rivals complex machine learning for chronic disease prediction (Nusinovici et al., 2020). Foundation models have been applied to structured prediction across domains, though their representation geometry, particularly the linear separability of clinical prediction targets, is not well characterized. Prior work on medical imaging fairness establishes that strong average performance can mask important subgroup disparities (Seyyed-Kalantari et al., 2021; Oakden-Rayner et al., 2020), a phenomenon that motivates per-demographic evaluation of FM-derived structured features. MedImageInsight (Codella et al., 2024) is pretrained across imaging modalities including CT; Med-SigLIP (Health AI Developer Foundations, 2025) aligns medical images and text, and BiomedCLIP (Zhang et al., 2024) is trained on 15M figure-caption pairs using a ViT-B/16 encoder. Multimodal fusion does not universally improve on single-modality pipelines (Krones et al., 2025; Wu et al., 2024), reinforcing the value of characterizing isolated visual embeddings.

## 3. Methods

**Dataset.** INSPECT (Huang et al., 2023) contains 23,248 CTPA studies from 19,402 unique patients with time-stamped structured EHR and curated PE diagnostic and prognostic labels (patient-level split 80%/5%/15%). Younger PE patients are rarer, may present atypically, and are underrepresented in training, contributing to age disparities independent of encoder limitations.

**Frozen Encoders as Structured Feature Extractors.** All models are frozen feature extractors without fine-tuning. CTPA scans are decomposed into axial slices, clipped to a lung-relevant Hounsfield Unit window, and normalized to 8-bit. Slice embeddings are mean-pooled uniformly across all models to produce fixed-length structured feature vectors. Mean pooling dilutes focal PE signals but transfers diffuse prognostic signals well. MedImageInsight (Codella et al., 2024) yields 1,024-d vectors (632M; ViT-H/14); MedSigLIP (Health AI Developer Foundations, 2025) yields 1,152-d vectors (400M; SigLIP ViT-So/14); BiomedCLIP (Zhang et al., 2024) yields 512-d vectors (86M; ViT-B/16), 4–7× smaller.

**Structured Prediction Probes.** We evaluate logistic regression (LR) as a linear probe and MLPs of varying depth on frozen structured embedding vectors; full MLP results appear in Appendix A. Hyperparameters are selected by grid search on validation AUROC: $C \in \{0.001, 0.01, 0.1, 1.0, 10.0\}$ for LR; hidden sizes (256, 128), (512, 256), (512, 256, 128) for MLPs.

**Multimodal Ablation.** To assess whether structured EHR can close representational gaps in frozen-encoder pipelines, we train LR probes on three input variants per encoder: CT-only (frozen embedding), EHR-only (structured features—demographics, vitals, standard labs—within a 24-hour window of the CTPA scan, mean-imputed), and CT+EHR (concatenated).

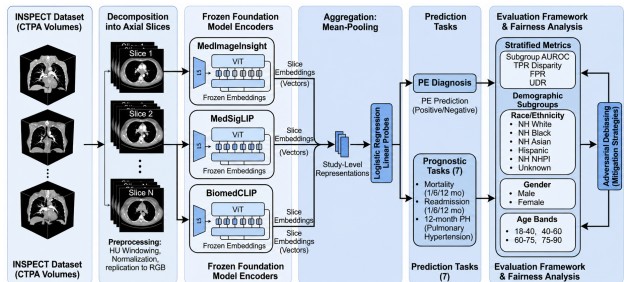

*Figure 1.* Pipeline overview. CTPA volumes are decomposed into axial slices and encoded with a frozen foundation model. Slice embeddings are mean-pooled into study-level representations and passed to a probe for PE prediction. A fairness evaluation reports per-subgroup AUROC, TPR disparity, FPR, and UDR.

**Evaluation.** AUROC with 95% CIs via 1,000 bootstrap samples; AUC differences via DeLong test (DeLong et al., 1988); TPR disparity significance via 1,000 permutation samples; adversarial debiasing significance: $p_{\text{gap}} = P(\text{bootstrapped baseline gap} \leq \text{post-debiasing gap})$.

**Fairness Metrics.** Per-subgroup AUROC, TPR, FPR, and UDR $(= 1 - \text{TPR})$. TPR disparity $= \text{TPR}_{group} - \text{TPR}_{reference}$ (NH White for race/ethnicity; Male for gender; no reference for age). Min–max UDR gap identifies within-category inequality. NH NHPI ($n=60$) yields CIs spanning $[0, 1]$ and is excluded from all fairness conclusions.

**Bias Mitigation.** (1) Importance weighting (IW): reweight by inverse group and positive class frequency. (2) Group resampling: equalize group sizes. (3) Adversarial debiasing: a gradient reversal layer penalises demographic retention with strength $\alpha \in \{0.5, 1.0, 2.0\}$ selected per protected attribute on the validation set. Reported gap reductions should be treated as upper bounds until replicated on held-out cohorts.

## 4. Results

### 4.1. Representation Geometry: Linear vs. MLP Probes

LR outperforms all MLP variants for MedImageInsight ($+0.014$) and MedSigLIP ($+0.021$), while BiomedCLIP shows a small MLP advantage ($+0.010$). This is consistent with approximately linearly separable clinical prediction targets in high-capacity encoder embeddings, where the

*Table 1.* PE diagnostic AUROC. MedImageInsight and MedSigLIP are statistically indistinguishable ($p$=0.952); all three fall significantly below the CT-LRCN baseline ($p \leq 0.009$). LR outperforms all MLP variants for MII and MSig (Appendix A).

| Model | AUROC | 95% CI | Params |
|---|---|---|---|
| CT-LRCN baseline | **0.721** | (0.69, 0.75) | 271M |
| MedImageInsight | 0.680 | (0.655, 0.706) | 632M |
| MedSigLIP | **0.684** | (0.660, 0.708) | 400M |
| BiomedCLIP | 0.626 | (0.599, 0.655) | 86M |

*Table 2.* UDR by demographic subgroup. **Our main finding: age is the dominant disparity dimension.** Age-band UDR gaps (0.32–0.45) exceed race/ethnicity (0.21–0.29) and gender (0.08–0.16) and persist across all eight structured prediction tasks. ${}^{**}p$<0.01, ${}^{*}p$<0.05, ${}^{a}p$>0.10 (permutation test, 1,000 samples). ${}^{\dagger}$NH NHPI excluded.

| Cat. | Subgroup | $n$ | MII UDR | MSig UDR | BCL UDR |
|---|---|---|---|---|---|
| Race | NH White (ref) | 1,608 | 0.474 | 0.459 | 0.378 |
| | NH Black | 189 | 0.419 | **0.613** | 0.387 |
| | NH Asian | 555 | **0.566** | 0.553 | **0.513** |
| | Hispanic${}^{a}$ | 497 | 0.552 | 0.552 | 0.478 |
| | *Min–max gap* | | **0.233** | **0.207** | **0.288** |
| Gender | Male (ref) | 1,369 | 0.430 | 0.449 | 0.325 |
| | Female${}^{**}$ | 1,841 | **0.556** | **0.531** | **0.486** |
| | *Min–max gap* | | **0.126** | **0.082** | **0.161** |
| Age | 18–40 | 359 | **0.743** | **0.800** | **0.629** |
| | 40–60 | 879 | 0.521 | 0.556 | 0.486 |
| | 60–75 | 1,093 | 0.548 | 0.538 | 0.472 |
| | 75–90 | 689 | 0.410 | 0.347 | 0.313 |
| | *Min–max gap* | | **0.333** | **0.453** | **0.316** |

smaller BiomedCLIP space benefits from nonlinear probing. These differences are small relative to overlapping AUROC CIs, however, so default escalation to MLPs may still underperform LR for reasons beyond dimensionality alone. The $\approx$4-point gap below CT-LRCN reflects dilution of the focal vascular signal by mean pooling. Strong prognostic performance (mortality AUROC 0.83–0.90; Appendix C) confirms that diffuse signals transfer well into structured feature vectors.

### 4.2. EHR Fusion Ablation

EHR alone scores 0.585 on PE diagnosis versus 0.680–0.684 for frozen CT alone; fusion adds at most +0.003 for high-capacity encoders and +0.017 for BiomedCLIP readmission (full breakdown in Appendix B). Gains are near-zero or negative for mortality. Adding EHR to frozen-encoder systems cannot compensate for representational deficits: the bottleneck is spatial aggregation, and structured EHR cannot recover lost focal vascular signal.

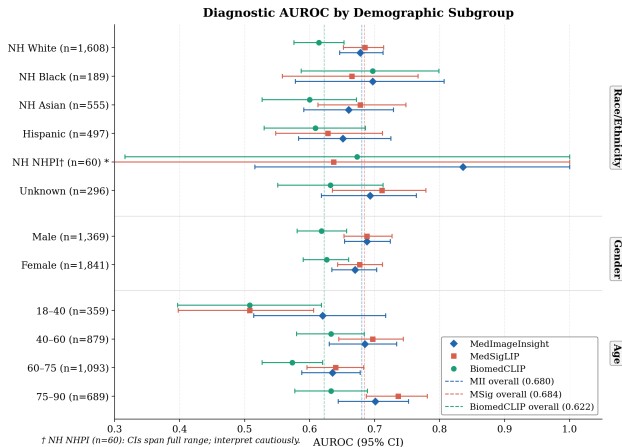

*Figure 2.* Diagnostic AUROC by demographic subgroup. The 18–40 age group reveals near-chance AUROC (0.508) for MedSigLIP and BiomedCLIP, a property of the embedding space, not probe choice. ${}^{\dagger}$NH NHPI ($n$=60): excluded from interpretation.

*Table 3.* Min–max UDR gaps across mitigation strategies. ${}^{***}p$<0.001, ${}^{**}p$<0.01, ${}^{*}p$<0.05, ${}^{\wedge}p$<0.10, ${}^{ns}p$≥0.10 (one-sided bootstrap). IW and group resampling in Appendix G.

| Model | Strategy | AUC | Race | Gender | Age |
|---|---|---|---|---|---|
| MII | Baseline | 0.680 | 0.233 | 0.126 | 0.333 |
| | Adv. race $\alpha$=2.0${}^{\wedge}$ | 0.673 | **0.117** | **0.071** | **0.199** |
| | Adv. gender $\alpha$=0.5${}^{*}$ | **0.680** | 0.493 | 0.044 | 0.268 |
| | Adv. age $\alpha$=0.5${}^{***}$ | 0.669 | 0.316 | 0.133 | **0.069** |
| MSig | Baseline | 0.684 | 0.207 | 0.082 | 0.453 |
| | Adv. race $\alpha$=0.5${}^{ns}$ | 0.666 | 0.215 | 0.015 | 0.434 |
| | Adv. gender $\alpha$=0.5${}^{***}$ | 0.665 | 0.247 | **0.000** | 0.490 |
| | Adv. age $\alpha$=0.5${}^{**}$ | 0.663 | 0.376 | 0.105 | **0.253** |
| BCL | Baseline | 0.626 | 0.288 | 0.161 | 0.316 |
| | Best adv.${}^{ns}$ (all attrs.) | 0.618–0.632 | No significant reduction ($p$≥0.12) | | |

### 4.3. Fairness Audit of Structured Representations

All three encoders fail systematically for patients aged 18–40: MedSigLIP and BiomedCLIP reach near-chance AUROC (0.508), and MedImageInsight achieves only 0.620 (per-encoder age AUROC in Appendix E). Table 2 quantifies UDR gaps of 0.32–0.45 ($p$<0.001), consistent across all seven prognostic tasks (0.04–0.20; Appendix K) and in the CT-LRCN baseline (UDR = 0.571; Appendix J). Female TPR disparity is significant across all three models ($p$≤0.040) and persists at all thresholds (Appendix F), ruling out a threshold artifact. The CT-LRCN baseline reverses the gender pattern (Female TPR = 0.592 > Male), suggesting that frozen-encoder pipelines encode gender differently from end-to-end trained representations.

### 4.4. Post-hoc Debiasing of the Structured Prediction Pipeline

Only adversarial debiasing reduces demographic gaps without substantially hurting AUROC. Importance weighting widens race/ethnicity gaps and group resampling collapses

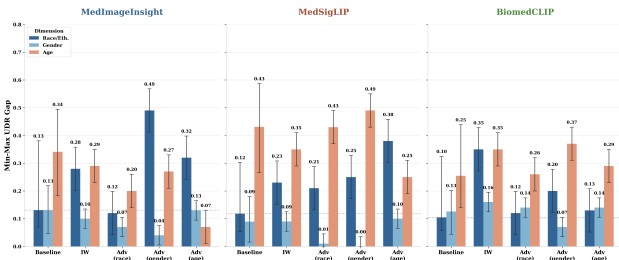

*Figure 3.* Min–max UDR gap by mitigation strategy. Age-targeted adversarial debiasing ($\alpha{=}0.5$, $p{<}0.001$) achieves the single largest reduction for MedImageInsight ($0.333{\rightarrow}0.069$). BiomedCLIP yields no statistically significant reductions, and group resampling collapses AUROC to $\approx 0.56$.

task performance to $\approx 0.55$–$0.56$ (Appendix G). Neither is recommended. For MedImageInsight, age-targeted debiasing reduces the age UDR gap by 79% ($0.333{\rightarrow}0.069$; $p{<}0.001$) at 0.011 AUROC cost. Race-targeted debiasing ($\alpha{=}2.0$) also reduces gender and age gaps concurrently (50%, 44%, 40% respectively) at only 0.007 AUROC cost ($p{=}0.061$; suggestive but non-significant at $\alpha{=}0.05$), consistent with partially shared demographic structure in the 1,024-d space. For MedSigLIP, gender-targeted ($\alpha{=}0.5$; $p{<}0.001$) eliminates the gender gap entirely ($0.082{\rightarrow}0.000$), and age-targeted yields a 44% reduction ($p{=}0.007$). BiomedCLIP shows numerically meaningful reductions (up to 60% for race) but none are statistically significant ($p{\geq}0.12$). BiomedCLIP's 512-d space encodes demographics less separably (Appendix L), suggesting that gradient reversal requires a minimum demographic separability.

## 5. Discussion

Embedding geometry matters: LR outperforms MLPs for MedImageInsight and MedSigLIP, indicating linearly separable clinical targets in high-capacity spaces. Structured EHR fusion adds negligible value over frozen CT embeddings alone (at most +0.017 AUROC): the bottleneck is spatial aggregation, and EHR cannot recover focal vascular signal lost to mean-pooling. Structured clinical pipelines should prioritize attention-based slice pooling or lightweight fine-tuning over fusion architectures.

The more consequential finding is the age-band disparity. UDR gaps of 0.32–0.45 for patients aged 18–40 versus 75–90 persist across all eight tasks and in the CT-LRCN baseline, implicating data distribution rather than encoder design and motivating targeted augmentation or oversampling for younger patients alongside post-hoc debiasing. At the same time, the frozen-encoder paradigm appears to introduce its own fairness costs not present in end-to-end trained models: the CT-LRCN baseline reverses the gender disparity (Female TPR $=0.592 >$ Male), while frozen encoders

systematically disadvantage female patients at all thresholds, a gap invisible to aggregate evaluation. Race-targeted debiasing for MedImageInsight concurrently reduces gender and age gaps, an empirical pattern consistent with partially shared demographic latents in the 1,024-d space, warranting further study.

Age is a required fairness axis for structured clinical AI: the near-chance AUROC for the 18–40 group makes omitting age from subgroup evaluation a patient-safety blind spot, and per-group reporting across race, gender, and age should be a minimum standard for FM-derived pipelines.

### 5.1. Limitations

First, all encoder comparisons rely on a single frozen slice-wise mean-pooling pipeline. We did not ablate alternative visual aggregation strategies such as attention-pooling or max-pooling (Huang et al., 2023). A portion of the diagnostic gap relative to CT-LRCN, and the demographic patterns themselves, may thus reflect this feature engineering choice rather than fundamental embedding limitations. Second, while the persistent failure mode in the 18–40 age cohort across all models and the baseline suggests a shared data-distributional bottleneck, we have not directly isolated the precise clinical mechanisms (e.g., via stratified label-quality analysis or chart review of atypical presentations). This near-chance diagnostic performance is also heavily confounded by the extremely low baseline prevalence of positive pulmonary embolism cases and small sample sizes in the younger cohort, so the observed age disparity is only partially attributable to representation-level biases. Third, all findings are restricted to binary tasks on a single CTPA benchmark requiring multi-center validation, and reported debiasing gap reductions are upper bounds pending held-out replication.

## 6. Conclusion

We characterize three medical imaging foundation models as structured feature extractors on a multimodal clinical benchmark. High-capacity embeddings exhibit linear geometry, favoring LR over reflexive MLP escalation, and EHR fusion cannot compensate for their representational gaps. Most consequentially, age-band disparities dominate race and gender effects across all eight tasks, with near-chance AUROC for patients aged 18–40 in two of three encoders. Age-targeted adversarial debiasing reliably mitigates this for high-capacity encoders but not BiomedCLIP, whose smaller embedding space offers insufficient demographic separability. These results establish per-group age reporting as a requirement, not an option, for responsible benchmarking of FM-derived structured representations in clinical deployment.

## Impact Statement

This paper reveals demographic disparities in medical imaging FM embeddings on a multimodal clinical benchmark. Age-band UDR gaps of 0.32–0.45 with near-chance diagnostic AUROC for patients aged 18–40, persistent across all eight tasks, have direct implications for FM-derived structured feature pipelines in clinical deployment. Age-targeted adversarial debiasing is a reliable post-hoc mitigation for high-capacity encoders; smaller encoders such as Biomed-CLIP should not be deployed for fairness-critical tasks without further validation. All experiments use the public IN-SPECT dataset; findings are institution-specific and require multi-center validation.

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

# Appendices

## A. MLP Probe Results

*Table 4.* Diagnostic AUROC for LR vs. best MLP probe. LR outperforms all MLP variants for MedImageInsight (+0.014) and MedSigLIP (+0.021), consistent with approximately linearly separable task-relevant structure in high-capacity encoder embeddings. BiomedCLIP MLP-small exceeds LR by 0.010. All MLPs: Adam, batch norm, dropout 0.2, early stopping (patience = 20), max 300 epochs.

| Model | Probe | Hidden Sizes | AUROC | 95% CI |
|---|---|---|---|---|
| CT-LRCN baseline | – | – | **0.721** | (0.69, 0.75) |
| MedImageInsight | **LR** | – | **0.680** | (0.655, 0.706) |
| MedImageInsight | MLP-best | (512,256,128) | 0.662 | (0.639, 0.687) |
| MedSigLIP | **LR** | – | **0.684** | (0.660, 0.708) |
| MedSigLIP | MLP-best | (512,256) | 0.656 | (0.634, 0.681) |
| BiomedCLIP | LR | – | 0.626 | (0.599, 0.655) |
| BiomedCLIP | **MLP-best** | (256,128) | **0.634** | (0.611, 0.661) |

## B. Full EHR Fusion Ablation

*Table 5.* Full CT-only vs. EHR-only vs. CT+EHR ablation (LR probe) across all tasks and encoders. EHR alone is substantially weaker than CT on every task. Fusion gains are marginal or negative for high-capacity encoders; the largest gains occur for BiomedCLIP readmission (+0.013–+0.017). $\Delta$ = Fusion − CT-only. **Bold** marks best per row.

| Model | Task | CT | EHR | Fusion | $\Delta$ |
|---|---|---|---|---|---|
| MedSigLIP | PE diagnosis | 0.684 | 0.585 | **0.687** | +0.003 |
| | 1-mo mortality | **0.881** | 0.601 | 0.879 | −0.002 |
| | 6-mo mortality | **0.844** | 0.593 | **0.844** | ±0.000 |
| | 12-mo mortality | **0.825** | 0.595 | **0.825** | +0.001 |
| | 1-mo readmission | **0.616** | 0.570 | 0.611 | −0.005 |
| | 6-mo readmission | 0.629 | 0.573 | **0.633** | +0.004 |
| | 12-mo readmission | 0.655 | 0.580 | **0.660** | +0.005 |
| BiomedCLIP | PE diagnosis | 0.626 | 0.585 | **0.634** | +0.008 |
| | 1-mo mortality | **0.865** | 0.601 | **0.865** | −0.000 |
| | 6-mo mortality | **0.820** | 0.593 | **0.821** | +0.000 |
| | 12-mo mortality | 0.807 | 0.595 | **0.809** | +0.002 |
| | 1-mo readmission | 0.581 | 0.570 | **0.594** | +0.013 |
| | 6-mo readmission | 0.598 | 0.573 | **0.614** | +0.017 |
| | 12-mo readmission | 0.606 | 0.580 | **0.622** | +0.016 |
| MedImageInsight | PE diagnosis | 0.680 | 0.585 | **0.682** | +0.002 |
| | 1-mo mortality | **0.891** | 0.601 | **0.891** | −0.000 |
| | 6-mo mortality | **0.846** | 0.593 | **0.847** | +0.001 |
| | 12-mo mortality | **0.830** | 0.595 | **0.831** | +0.001 |
| | 1-mo readmission | **0.622** | 0.570 | 0.616 | −0.006 |
| | 6-mo readmission | 0.632 | 0.573 | **0.639** | +0.007 |
| | 12-mo readmission | 0.632 | 0.580 | **0.641** | +0.009 |

## C. Full Prognostic Performance

*Table 6.* Diagnostic and prognostic AUROC and ECE. Strong mortality prediction (AUROC 0.83–0.90) confirms diffuse CT signals encode well into FM structured feature vectors. **Bold** marks best AUROC per task.

| Task | MedImageInsight AUROC (95% CI) | ECE | MedSigLIP AUROC (95% CI) | ECE | BiomedCLIP AUROC (95% CI) | ECE |
|---|---|---|---|---|---|---|
| PE Diagnosis | 0.680 (0.655–0.706) | 0.044 | **0.684 (0.660–0.708)** | 0.041 | 0.626 (0.599–0.655) | **0.035** |
| 1-mo Mortality | **0.898 (0.875–0.917)** | 0.014 | 0.890 (0.863–0.908) | 0.016 | 0.869 (0.842–0.894) | **0.013** |
| 6-mo Mortality | **0.856 (0.835–0.875)** | 0.019 | 0.851 (0.832–0.870) | **0.140** | 0.835 (0.812–0.856) | 0.020 |
| 12-mo Mortality | **0.847 (0.827–0.865)** | **0.014** | 0.844 (0.823–0.865) | 0.014 | 0.825 (0.805–0.845) | 0.018 |
| 1-mo Readmit | **0.616 (0.563–0.663)** | 0.023 | 0.585 (0.539–0.637) | 0.023 | 0.580 (0.524–0.630) | **0.006** |
| 6-mo Readmit | **0.645 (0.614–0.677)** | **0.011** | 0.622 (0.587–0.654) | 0.050 | 0.604 (0.570–0.639) | 0.018 |
| 12-mo Readmit | **0.651 (0.623–0.678)** | **0.007** | 0.649 (0.622–0.678) | 0.027 | 0.618 (0.590–0.646) | 0.009 |
| 12-mo PH | **0.755 (0.728–0.781)** | 0.017 | 0.753 (0.726–0.778) | **0.014** | 0.711 (0.683–0.739) | 0.013 |

## D. Full TPR, FPR, and UDR Tables

*Table 7.* Full per-subgroup metrics for MedImageInsight (LR, threshold = 0.235). [†]NH NHPI ($n$=60) excluded. **Bold** marks worst per category.

| Category | Subgroup | $n$ | TPR (95% CI) | FPR (95% CI) | UDR |
|---|---|---|---|---|---|
| Race/Eth. | NH White (ref) | 1,608 | 0.526 (0.473–0.582) | 0.261 (0.237–0.287) | 0.474 |
| | NH Black | 189 | 0.581 (0.414–0.750) | 0.184 (0.126–0.247) | 0.419 |
| | NH Asian | 555 | **0.434 (0.325–0.547)** | 0.203 (0.171–0.239) | **0.566** |
| | Hispanic | 497 | 0.448 (0.338–0.571) | 0.167 (0.131–0.203) | 0.552 |
| | NH NHPI[†] | 60 | 0.667 (0.000–1.000) | **0.123 (0.036–0.214)** | 0.333 |
| | Unknown | 296 | 0.476 (0.358–0.596) | 0.236 (0.185–0.291) | 0.524 |
| Gender | Male (ref) | 1,369 | 0.570 (0.511–0.632) | 0.273 (0.246–0.298) | 0.430 |
| | Female | 1,841 | **0.444 (0.388–0.495)** | **0.192 (0.173–0.212)** | **0.556** |
| Age | 18–40 | 359 | **0.257 (0.114–0.419)** | **0.142 (0.104–0.182)** | **0.743** |
| | 40–60 | 879 | 0.479 (0.393–0.563) | 0.181 (0.156–0.208) | 0.521 |
| | 60–75 | 1,093 | 0.452 (0.384–0.522) | 0.251 (0.223–0.283) | 0.548 |
| | 75–90 | 689 | 0.590 (0.508–0.673) | 0.270 (0.230–0.308) | 0.410 |

*Table 8.* Full per-subgroup TPR, FPR, and UDR for MedSigLIP (threshold = 0.240) and BiomedCLIP (threshold = 0.192). [†]NH NHPI excluded. **Bold** marks worst per category.

| Category | Subgroup | $n$ | MSig TPR (95% CI) | MSig UDR | BCL TPR (95% CI) | BCL UDR |
|---|---|---|---|---|---|---|
| Race/Eth. | NH White (ref) | 1,608 | 0.541 (0.489–0.589) | 0.459 | 0.622 (0.566–0.672) | 0.378 |
| | NH Black | 189 | **0.387 (0.226–0.556)** | **0.613** | 0.613 (0.446–0.791) | 0.387 |
| | NH Asian | 555 | 0.447 (0.342–0.561) | 0.553 | **0.487 (0.372–0.607)** | **0.513** |
| | Hispanic | 497 | 0.448 (0.329–0.578) | 0.552 | 0.522 (0.406–0.635) | 0.478 |
| | NH NHPI[†] | 60 | 0.333 (0.000–1.000) | 0.667 | 0.333 (0.000–1.000) | 0.667 |
| | Unknown | 296 | 0.540 (0.412–0.648) | 0.460 | 0.619 (0.509–0.729) | 0.381 |
| Gender | Male (ref) | 1,369 | 0.551 (0.494–0.613) | 0.449 | 0.676 (0.621–0.733) | 0.325 |
| | Female | 1,841 | **0.469 (0.412–0.525)** | **0.531** | **0.515 (0.462–0.571)** | **0.486** |
| Age | 18–40 | 359 | **0.200 (0.075–0.344)** | **0.800** | **0.371 (0.203–0.543)** | **0.629** |
| | 40–60 | 879 | 0.444 (0.362–0.528) | 0.556 | 0.514 (0.438–0.602) | 0.486 |
| | 60–75 | 1,093 | 0.462 (0.398–0.532) | 0.538 | 0.528 (0.461–0.598) | 0.472 |
| | 75–90 | 689 | 0.653 (0.570–0.730) | 0.347 | 0.688 (0.605–0.772) | 0.313 |

## E. Diagnostic AUROC by Age Group

*Table 9.* Dataset demographic distribution and positive case counts. The extremely low base rate of positive PE cases in the 18–40 cohort acts as a significant confounder for diagnostic performance.

| Subgroup | Total Patients ($n$) | Positive Cases | Prevalence (%) |
|---|---|---|---|
| Age 18–40 | 359 | 35 | 9.75 |
| Age 40–60 | 879 | 144 | 16.38 |
| Age 60–75 | 1,093 | 199 | 18.21 |
| Age 75–90 | 689 | 144 | 20.91 |

## F. Threshold Analysis

*Table 10.* Gender-targeted threshold analysis. Female disparity persists for all three foundation models at all thresholds—a representation-level property, not a threshold artifact. CT-LRCN reverses the disparity direction (Female Disp = +0.036). **Bold** marks smaller absolute Female Disp.

| Model | Threshold | Female TPR | Male TPR | Female Disp |
|---|---|---|---|---|
| MedImageInsight | 0.103 | 0.752 | 0.857 | −0.104 |
| MedSigLIP | 0.112 | 0.762 | 0.845 | **−0.083** |
| BiomedCLIP | 0.137 | 0.736 | 0.875 | −0.139 |
| CT-LRCN | 0.090 | **0.817** | 0.781 | +0.036 |

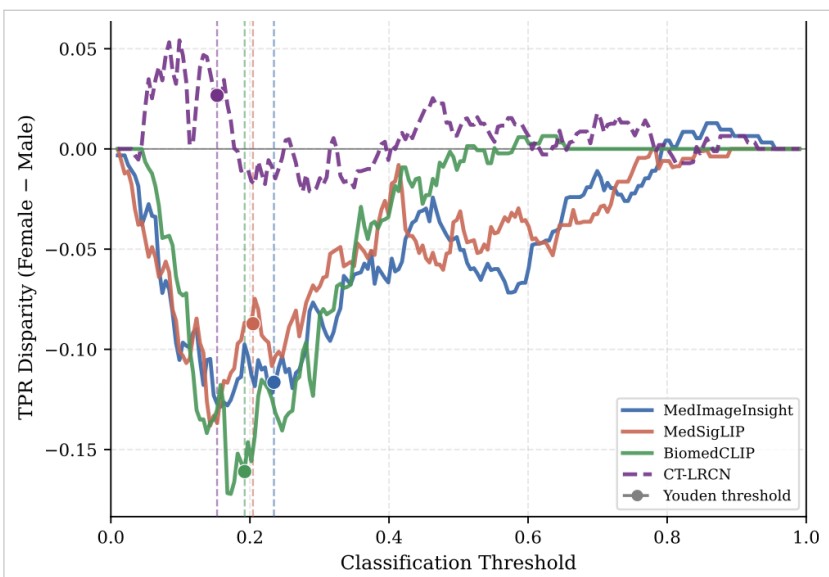

*Figure 4.* TPR disparity vs. classification threshold for Female vs. Male. Dashed vertical lines mark the global Youden threshold per model. The female disparity remains near-constant across all thresholds, confirming a representation-level gap independent of threshold choice.

## G. Importance Weighting and Group Resampling

Importance weighting widened race/ethnicity UDR gaps for all three models (MII: 0.233 → 0.279; MSig: 0.207 → 0.233; BCL: 0.288 → 0.351) and produced opposing effects on Hispanic disparity across models. Group resampling collapsed AUROC to ≈0.55–0.56 while dramatically widening race gaps for MedImageInsight (0.233→0.434) and MedSigLIP (0.207→0.582). NH NHPI achieves TPR = 1.000 under MII resampling, an extreme oversampling artifact ($n$=60). Neither IW nor group resampling is recommended as a post-hoc debiasing strategy for FM-derived structured prediction pipelines.

# H. Stability Analysis

*Table 11.* AUROC stability across 10 bootstrap resamples of the training set. Std < 0.01 for all three models.

| Model | Mean AUROC | Std AUROC |
| --- | --- | --- |
| MedImageInsight | **0.666** | **0.005** |
| MedSigLIP | **0.666** | 0.007 |
| BiomedCLIP | 0.618 | 0.006 |

*Table 12.* Per-subgroup TPR stability (mean ± std) across 10 bootstrap resamples. The 18–40 age group shows the highest variance, consistent with small sample size and low TPR. **Bold** marks worst (lowest) mean TPR per category.

| Category | Subgroup | MII TPR (mean ± std) | MSig TPR (mean ± std) | BCL TPR (mean ± std) |
| --- | --- | --- | --- | --- |
| Race/Eth. | NH White | $0.555 \pm 0.065$ | $0.638 \pm 0.061$ | $0.545 \pm 0.088$ |
| | NH Black | $0.568 \pm 0.073$ | $0.629 \pm 0.100$ | $0.574 \pm 0.101$ |
| | NH Asian | $0.465 \pm 0.065$ | $0.550 \pm 0.061$ | $0.455 \pm 0.090$ |
| | Hispanic | $\mathbf{0.427 \pm 0.042}$ | $\mathbf{0.521 \pm 0.075}$ | $\mathbf{0.461 \pm 0.138}$ |
| | Unknown | $0.541 \pm 0.084$ | $0.652 \pm 0.065$ | $0.552 \pm 0.079$ |
| Gender | Male | $0.579 \pm 0.071$ | $0.662 \pm 0.068$ | $0.590 \pm 0.096$ |
| | Female | $\mathbf{0.482 \pm 0.053}$ | $\mathbf{0.570 \pm 0.056}$ | $\mathbf{0.469 \pm 0.083}$ |
| Age | 18–40 | $\mathbf{0.311 \pm 0.058}$ | $\mathbf{0.269 \pm 0.068}$ | $\mathbf{0.291 \pm 0.129}$ |
| | 40–60 | $0.494 \pm 0.053$ | $0.572 \pm 0.062$ | $0.442 \pm 0.094$ |
| | 60–75 | $0.480 \pm 0.060$ | $0.593 \pm 0.067$ | $0.483 \pm 0.084$ |
| | 75–90 | $0.623 \pm 0.070$ | $0.709 \pm 0.064$ | $0.620 \pm 0.100$ |

# I. Calibration (ECE)

*Table 13.* Expected Calibration Error (ECE, 10 equal-width bins) with 95% bootstrap CIs. BiomedCLIP's lower ECE reflects broader probability coverage under its lower Youden threshold (0.192). **Bold** marks best (lowest) ECE.

| Model | ECE | 95% CI |
| --- | --- | --- |
| MedImageInsight (LR) | 0.044 | (0.035, 0.058) |
| MedSigLIP (LR) | 0.041 | (0.032, 0.055) |
| BiomedCLIP (LR) | **0.035** | (0.024, 0.047) |

## J. CT-LRCN Baseline Fairness (PE Diagnosis)

*Table 14.* Per-subgroup TPR, FPR, and UDR for the INSPECT CT-LRCN baseline (Youden threshold; AUROC = 0.720). Unlike all three foundation models, the baseline shows Female TPR = 0.592 > Male TPR = 0.558. [†]NH NHPI excluded. **Bold** marks worst per category.

| Category | Subgroup | $n$ | TPR (95% CI) | FPR (95% CI) | UDR |
|---|---|---|---|---|---|
| Race/Eth. | NH White (ref) | 1,608 | 0.619 (0.566–0.668) | 0.253 (0.229–0.276) | 0.381 |
| | NH Black | 189 | 0.677 (0.500–0.833) | 0.209 (0.148–0.272) | 0.323 |
| | NH Asian | 555 | 0.539 (0.422–0.648) | 0.265 (0.227–0.307) | 0.461 |
| | Hispanic | 497 | **0.403 (0.288–0.523)** | 0.214 (0.174–0.254) | **0.597** |
| | NH NHPI[†] | 60 | 0.667 (0.000–1.000) | 0.140 (0.054–0.246) | 0.333 |
| | Unknown | 296 | 0.508 (0.396–0.627) | 0.223 (0.172–0.276) | 0.492 |
| Gender | Male (ref) | 1,369 | 0.558 (0.496–0.619) | 0.235 (0.212–0.261) | 0.442 |
| | Female | 1,841 | **0.592 (0.537–0.644)** | 0.244 (0.223–0.265) | **0.408** |
| Age | 18–40 | 359 | **0.429 (0.259–0.600)** | 0.182 (0.142–0.226) | **0.571** |
| | 40–60 | 879 | 0.535 (0.450–0.615) | 0.196 (0.168–0.224) | 0.465 |
| | 60–75 | 1,093 | 0.578 (0.513–0.651) | 0.243 (0.213–0.272) | 0.422 |
| | 75–90 | 689 | 0.597 (0.520–0.674) | 0.295 (0.258–0.334) | 0.403 |

## K. Prognostic Fairness

*Table 15.* Min–max UDR gaps across all seven prognostic tasks. Age-band gaps (0.04–0.20) are consistently present across all tasks and models. [†]CT-LRCN mortality race gaps inflated by NH NHPI. [‡]1-month readmission race gaps for FMs inflated by NH NHPI ($n<60$). **Bold** marks largest gap per task per demographic dimension.

| Task | MedImageInsight | | | MedSigLIP | | | BiomedCLIP | | | CT-LRCN | | |
|---|---|---|---|---|---|---|---|---|---|---|---|---|
| | R/E | Gen | Age | R/E | Gen | Age | R/E | Gen | Age | R/E | Gen | Age |
| 1-mo mortality | 0.288 | 0.028 | 0.099 | **0.426** | 0.043 | 0.083 | 0.246 | 0.052 | 0.062 | 0.258[†] | 0.008 | **0.134** |
| 6-mo mortality | 0.199 | 0.029 | 0.133 | 0.133 | 0.004 | **0.148** | 0.138 | 0.034 | 0.133 | 0.538[†] | 0.026 | 0.106 |
| 12-mo mortality | 0.268 | 0.007 | 0.093 | **0.425** | 0.011 | 0.094 | 0.155 | 0.045 | **0.125** | 0.438[†] | 0.002 | 0.046 |
| 1-mo readmission | 0.900[‡] | 0.029 | 0.165 | 0.750[‡] | 0.120 | **0.192** | 0.667[‡] | 0.138 | 0.125 | 0.349 | 0.029 | 0.144 |
| 6-mo readmission | 0.357 | 0.186 | 0.160 | 0.339 | 0.082 | 0.286 | 0.259 | 0.082 | **0.356** | 0.152 | **0.222** | 0.143 |
| 12-mo readmission | 0.318 | 0.116 | 0.159 | **0.405** | 0.155 | 0.257 | 0.233 | 0.156 | **0.343** | 0.302 | 0.140 | 0.042 |
| 12-mo PH | 0.278 | 0.005 | **0.218** | 0.167 | 0.047 | 0.174 | 0.097 | 0.018 | 0.103 | **0.342** | 0.054 | 0.196 |

## L. Demographic Probing

To investigate the embedding-capacity hypothesis for BiomedCLIP's unreliable debiasing, we trained LR and MLP classifiers on frozen embeddings to predict each demographic attribute. Chance-level macro-F1 is 0.250 for age band and race/ethnicity (four classes) and 0.500 for gender (two classes).

*Table 16.* Demographic probing macro-F1 (95% CI, 1,000 bootstrap samples). All values far exceed chance, confirming demographic signal is encoded in frozen structured representations. BiomedCLIP's markedly lower race/ethnicity F1 (0.537 vs. 0.677–0.707) indicates insufficient demographic separability for reliable gradient-reversal disentanglement. Negative MLP−LR gaps indicate linearly organised demographic structure, mirroring the LR≥MLP pattern on clinical tasks. **Bold** marks highest F1 per target and probe type.

| Probe | Target | MedImageInsight | MedSigLIP | BiomedCLIP |
|---|---|---|---|---|
| LR | Age band | **0.730** [0.713, 0.745] | 0.693 [0.674, 0.709] | 0.617 [0.600, 0.635] |
| | Gender | **0.972** [0.966, 0.977] | 0.966 [0.960, 0.972] | 0.939 [0.930, 0.947] |
| | Race/ethnicity | **0.707** [0.685, 0.728] | 0.677 [0.653, 0.698] | 0.537 [0.513, 0.560] |
| MLP | Age band | **0.705** [0.686, 0.722] | 0.678 [0.662, 0.696] | 0.589 [0.573, 0.607] |
| | Gender | **0.971** [0.965, 0.977] | 0.960 [0.954, 0.967] | 0.899 [0.889, 0.909] |
| | Race/ethnicity | **0.679** [0.657, 0.700] | 0.640 [0.618, 0.660] | 0.512 [0.491, 0.534] |
| MLP−LR gap | Age band | −0.025 | −0.015 | −0.028 |
| | Gender | −0.001 | −0.006 | −0.040 |
| | Race/ethnicity | −0.028 | −0.036 | −0.025 |
| Chance-level macro-F1 | | Age band: 0.250 | Gender: 0.500 | Race/ethnicity: 0.250 |

