# OpenReview forum: "Benchmarking Multimodal Clinical Foundation Models to Reveal Significant Demographic Disparities"
_ICML.cc/2026/Workshop/FMSD — FMSD @ ICML 2026 Poster_

### Official Review · Reviewer_2drd · 2026-05-14
**Submission 149 Review**

**Rating:** 6
**Confidence:** 4

**Review:**

## Summary

The paper presents an interesting perspective and analysis on the utilization of final-layer embedding vectors as structured tabular representations of medical biomarkers, and evaluates three medical foundation models across eight classification tasks. The study further provides an in-depth analysis of the fairness of these extracted embeddings across diverse demographic cohorts, while proposing strategies to mitigate bias for underrepresented groups. In addition, the authors investigate the integration of embedding-derived biomarkers with EHR data, offering empirical insights into scenarios where structured clinical information can complement the learned embedding representations, as well as scenarios where the addition of EHR data may provide limited or no further benefit.

## Strengths

- The use of foundation model embeddings as proxies for clinically relevant biomarkers is an active area of research within the medical community, and the paper further strengthens this direction by systematically evaluating the utility of 3 medical foundation model embeddings across 8 classification tasks.

- The paper is clearly written, develops strong intuition for the proposed analyses, and supports its claims with rigorous experimentation and in-depth discussion.

- The cohort-wise analysis across diverse demographic groups effectively highlights fairness concerns, which remain a critical challenge in medical foundation models. The findings could motivate future research toward developing more fair and unbiased healthcare foundation models through improved data selection, post-training adaptation, data curation, and evaluation paradigms.

- The paper demonstrates that EHR data is largely complementary and often provides only marginal gains over the core foundation model embeddings. It further discusses how strategies such as mean pooling may degrade fine-grained spatial information that is critical for localization-sensitive tasks. It also discusses alternatives such as attention-based slice pooling and lightweight fine-tuning over fusion architectures provides valuable practical insights for future multimodal medical AI systems.

- The paper empirically evaluates multiple debiasing strategies and provides detailed discussion regarding the scenarios in which each strategy is likely to be most effective.

## Weaknesses

- All evaluation experiments presented in the paper are limited to binary classification tasks. It would be valuable to investigate how the reported findings extend to multiclass classification settings and more complex clinical prediction scenarios.

- Furthermore, the evaluation is restricted to a single dataset. Since a significant portion of the paper’s claims are centered around fairness and demographic generalization, including diverse multi-center and multi-disease datasets would help strengthen and more comprehensively establish the paper’s conclusions.

- Since much of the discussion focuses on diverse demographic cohorts, the authors should consider including the demographic distribution of the dataset (possibly in the appendix) to help readers better understand the underlying data distribution and potential imbalance across cohorts. A more detailed discussion around these imbalances and their implications would also improve the paper.

- “EHR data” is a broad term, and the authors should explicitly specify which clinical features or modalities were included in the EHR representation to improve clarity and reproducibility.

## Suggestions

- While the paper is clearly written, it relies on several terminologies, evaluation strategies, and discussions that are more familiar to the medical AI community. Since this is a workshop focused on foundation models for structured data, it would be beneficial to include additional details in the form of equations, formulations, and clearer explanations of the evaluation methodologies to help readers from non-clinical backgrounds better engage with the paper.

- The authors should also consider providing expansions for abbreviations such as LRCN, along with appropriate references and discussion regarding why these models constitute relevant baselines for the task, in order to reduce ambiguity in interpretation.

- The authors demonstrate how embeddings of different dimensionalities, representative of varying encoder capacities, affect downstream performance. However, since these embeddings are derived from different architectures, it is difficult to attribute the observed trends solely to embedding dimensionality. Instead, the authors could project embeddings from a single model (e.g., MedSigLIP) into higher- and lower-dimensional representations and evaluate downstream performance on these projected embeddings to better support claims regarding representational capacity.

- The paper currently contains a limited number of references. Incorporating additional recent works related to medical foundation model embeddings, fairness analysis, multimodal fusion, and representation learning would further strengthen the paper. The following references may be particularly relevant and useful for the authors to consider.

Menon, T.P., Mahajan, A. & Powell, D. Foundation model embeddings for multimodal oncology data integration. npj Digit. Med. 9, 131 (2026). https://doi.org/10.1038/s41746-025-02312-8

Amar, Jonathan, et al. "Integrating Genomics into Multimodal EHR Foundation Models." arXiv preprint arXiv:2510.23639 (2025).

Cui, Saishi, et al. "Translating Histopathology Foundation Model Embeddings into Cellular and Molecular Features for Clinical Studies." bioRxiv (2026): 2026-03.

Lee, Yujin, et al. "Time-Series Foundation Model Embeddings as Means for Physiological Feature Extraction." 1st ICLR Workshop on Time Series in the Age of Large Models.

Mukherjee, Sumit, et al. "EmbedGEM: a framework to evaluate the utility of embeddings for genetic discovery." Bioinformatics Advances 4.1 (2024): vbae135.

Nusinovici, Simon, et al. "Logistic regression was as good as machine learning for predicting major chronic diseases." Journal of clinical epidemiology 122 (2020): 56-69.


## Justification of Score

The paper proposes a helpful evaluation and application of structured foundation models for a highly relevant downstream clinical task, while also highlighting important issues related to bias and fairness that may further guide the development of foundation models for clinical applications. The work could motivate interesting discussions at the workshop and encourage further research in this area. However, the paper would further benefit from stronger and more robust evaluation settings to better support the credibility and generalizability of its claims. Overall, the work would likely be of strong interest to the workshop audience and could encourage further research and engagement in this direction.

---

### Official Review · Reviewer_Dgu6 · 2026-05-20
**Critical Demographic Audit of Multimodal Clinical FMs: Linear Separability Assured, but Flawed Fusion and Opaque EHR Processing Limit Impact**

**Rating:** 5
**Confidence:** 5

**Review:**

**Page Limit Exceed Notice:** Please note that the manuscript slightly exceeds the strict 4-page content limit proposed by the conference track. Specifically, the final two lines of the "Limitations" section have spilled over onto the top of Page 5, immediately preceding the References. While this appears to be a minor LaTeX spacing/formatting issue rather than an intentional addition of core sections, I leave it to the Area Chairs' discretion to decide whether this requires administrative action or can be corrected by the authors during the camera-ready adjustments.

**Summary**

This paper presents a rigorous benchmark of three prominent multimodal clinical foundation models (*MedImageInsight*, *MedSigLIP*, and *BiomedCLIP*) on the public INSPECT dataset, which pairs computed tomography pulmonary angiography (CTPA) imaging with longitudinal Electronic Health Record (EHR) data. The authors evaluate these models across one diagnostic task (Pulmonary Embolism) and seven prognostic downstream tasks using both linear (Logistic Regression) and non-linear (MLP) probes. Their analysis yields three main findings: (1) simpler linear probes consistently outperform MLPs, indicating that high-capacity clinical embeddings are linearly separable; (2) multimodal fusion with structured EHR yields marginal performance gains; and (3) all models strongly encode latent demographic attributes (age, gender, race/ethnicity), leading to severe performance disparities that common bias-mitigation techniques  fail to decouple.

**Strengths**
*  The paper directly addresses the integration of multimodal foundation model representations with structured clinical tabular data (EHR), fitting perfectly within the scope of the FMSD workshop.
* Demonstrating that Logistic Regression systematically outperforms MLP variants is a significant empirical finding. It suggests that advanced clinical foundation models naturally organize downstream features into a linearly separable structure, heavily simplifying clinical deployment requirements.
* Rather than focusing solely on raw performance, the paper provides an essential ethical assessment, revealing that frozen high-capacity embeddings retain demographics.

**Areas for Improvement**
*  Although the title refers broadly to “Clinical Foundation Models”, the experiments only use one imaging modality (CTPA) and tasks related to Pulmonary Embolism in the INSPECT dataset. Because the evaluation is limited to this narrow setting, it is difficult to generalize the conclusions about representation limitations and demographic biases to medical imaging more broadly.
* The pipeline for processing the Electronic Health Records (EHR) is treated superficially in the manuscript. The authors mention utilizing structured data but fail to specify feature extraction protocols, normalization methods, temporal windows, or data granularity. Given that the EHR data yielded a negligible contribution to the overall results ($\le +0.003$ AUROC), the lack of transparency makes it impossible to determine whether this poor performance stems from a true lack of informational value in the clinical records or from inadequate preprocessing, unaddressed missing data, or lack of tabular harmonization.
*  The authors employ a rudimentary early-fusion mechanism (simple concatenation of frozen visual embeddings and tabular EHR features). Based on this trivial baseline, they heavily generalize that "fusion cannot compensate for representational deficits." This methodological choice is a major bottleneck; the simple concatenation may fail to capture complex non-linear interdependencies between modalities. Without exploring more sophisticated architectures (such as cross-attention modules, or end-to-end fine-tuning of the upper classification layers), the conclusion that multimodal fusion is ineffective remains unsubstantiated.

**Detailed Comments**
*  Please provide explicit details regarding the EHR processing pipeline. What was the exact feature dimensionality? How were missing values, which are notoriously prevalent in the INSPECT dataset, imputed or handled? What specific temporal window relative to the CTPA scan was considered for aggregating the longitudinal records?
* Did the authors consider utilizing a specialized tabular or clinical foundation model (e.g., TabPFN or a BERT-based EHR encoder) to generate high-capacity embeddings for the patient’s longitudinal record prior to fusion? Merging a highly contextualized clinical tabular embedding with the visual embedding would likely provide a more balanced cross-modal representation than concatenating raw features with frozen image vectors
* To elevate the paper's impact, could the authors replicate the demographic probing experiments on a more widely adopted multimodal dataset (such as MIMIC-CXR with MIMIC-IV tabular data) to prove that the failure of gradient-reversal debiasing is a universal property of these medical foundation models?


**Justification of Score**
The paper addresses an important topic by evaluating multimodal clinical foundation models and analyzing demographic biases in medical embeddings. The experiments are well-structured, and the strong performance of linear probes is an interesting finding. However, the EHR modality is underexplored, with limited methodological detail and a simplistic fusion strategy that weakens some of the broader claims about multimodal learning. Overall, the paper provides valuable observations but requires stronger validation and more rigorous multimodal analysis to fully support its conclusions.

---

### Official Review · Reviewer_M443 · 2026-05-20
**Review for "Benchmarking Multimodal Clinical Foundation Models to Reveal Significant Demographic Disparities"**

**Rating:** 7
**Confidence:** 3

**Review:**

# Summary

The paper treats three frozen medical imaging FM embeddings (MedImageInsight, MedSigLIP, BiomedCLIP) as structured feature vectors and benchmarks them on INSPECT across one PE diagnostic and seven prognostic tasks. It reports that linear probes match or beat MLPs on the high-capacity encoders, that EHR fusion adds almost nothing over frozen CT, and that age is the dominant disparity, with the 18-40 cohort near chance and UDR gaps of 0.32 to 0.45. Adversarial debiasing is the only mitigation that shrinks gaps without hurting AUROC, working on the larger encoders but failing on BiomedCLIP.

# Strengths

- The fairness audit is statistically careful, with bootstrap CIs, DeLong and permutation tests, and explicit p-values throughout.

- The paper honestly reports negative and non-significant results, including the failed BiomedCLIP debiasing and the age gap also appearing in the baseline.

- The age disparity is large, consistent across all eight tasks, and clinically consequential.

- The demographic-probing and stability appendices give the claims real backing beyond what a short paper usually offers.

# Areas for Improvement

- The headline age gap is confounded with base rate and small sample size, since the youngest cohort has few positives and very wide CIs and the gap also shows in the baseline.

- The probe-geometry finding rests on AUROC differences whose CIs overlap, with no significance test for LR versus MLP.

- The fusion conclusion conflates the chosen mean-pooling with representation quality, which the paper itself attributes to spatial aggregation.

- No multiple-comparison correction is applied across the many tasks, attributes, and α values tested.

# Detailed Comments

- Reporting per-band positive counts and base-rate-adjusted disparities would clarify how much of the 18-40 gap is a genuine embedding property.

- Debiasing reductions are validation-selected upper bounds, so a held-out or cross-cohort number would make the third finding more durable.

- An attention-pooled variant would separate aggregation effects from representational ones and sharpen the fusion claim.

# Justification of Score

This is a careful, honest, and statistically well-executed fairness audit with a clinically useful headline. The main weakness is interpretive: the central age disparity would benefit from base-rate-adjusted framing, and the probe-geometry and capacity claims should be softened. These are addressable framing issues rather than execution failures.